# Assessing Procedural Accuracy in Lateral Spine Surgery: A Retrospective Analysis of Percutaneous Pedicle Screw Placement with Intraoperative CT Navigation

**DOI:** 10.3390/jcm12216914

**Published:** 2023-11-03

**Authors:** Akihiko Hiyama, Daisuke Sakai, Hiroyuki Katoh, Satoshi Nomura, Masahiko Watanabe

**Affiliations:** Department Orthopaedic Surgery, Tokai University School of Medicine, 143 Shimokasuya, Isehara 259-1193, Japan; daisakai@is.icc.u-tokai.ac.jp (D.S.); hero@tokai-u.jp (H.K.); nomura.s@tokai-u.jp (S.N.); masahiko@is.icc.u-tokai.ac.jp (M.W.)

**Keywords:** percutaneous pedicle screws, intraoperative CT navigation, procedural accuracy, lateral spine surgery, spinal fusion, lateral lumbar interbody fusion

## Abstract

Percutaneous pedicle screws (PPSs) are commonly used in posterior spinal fusion to treat spine conditions such as trauma, tumors, and degenerative diseases. Precise PPS placement is essential in preventing neurological complications and improving patient outcomes. Recent studies have suggested that intraoperative computed tomography (CT) navigation can reduce the dependence on extensive surgical expertise for achieving accurate PPS placement. However, more comprehensive documentation is needed regarding the procedural accuracy of lateral spine surgery (LSS). In this retrospective study, we investigated patients who underwent posterior instrumentation with PPSs in the thoracic to lumbar spine, utilizing an intraoperative CT navigation system, between April 2019 and September 2023. The system’s methodology involved real-time CT-based guidance during PPS placement, ensuring precision. Our study included 170 patients (151 undergoing LLIF procedures and 19 trauma patients), resulting in 836 PPS placements. The overall PPS deviation rate, assessed using the Ravi scale, was 2.5%, with a notably higher incidence of deviations observed in the thoracic spine (7.4%) compared to the lumbar spine (1.9%). Interestingly, we found no statistically significant difference in screw deviation rates between upside and downside PPS placements. Regarding perioperative complications, three patients experienced issues related to intraoperative CT navigation. The observed higher rate of inaccuracies in the thoracic spine suggests that various factors may contribute to these differences in accuracy, including screw size and anatomical variations. Further research is required to refine PPS insertion techniques, particularly in the context of LSS. In conclusion, this retrospective study sheds light on the challenges associated with achieving precise PPS placement in the lateral decubitus position, with a significantly higher deviation rate observed in the thoracic spine compared to the lumbar spine. This study emphasizes the need for ongoing research to improve PPS insertion techniques, leading to enhanced patient outcomes in spine surgery.

## 1. Introduction

Posterior spinal fusion, with instrumentation using percutaneous pedicle screws (PPSs), has become a widely accepted surgical procedure for treating various spine diseases such as trauma, tumor, and degenerative disease [1,2,3]. Accurate PPS placement is critical, as misplacement can lead to serious complications like neurovascular injury and can impact a patient’s quality of life [4].

Traditionally, precise PPS placement is achieved using fluoroscopy, requiring significant surgical skill and experience, especially in insertion techniques.

However, recent studies have suggested that intraoperative computed tomography (CT) navigation may mitigate the need for extensive surgical experience in achieving precise placement [5,6,7]. In procedures involving lateral interbody fusion (LLIF) and PPS fixation, intraoperative repositioning is often required to facilitate easier PPS insertion. While there is a growing body of research on lateral spine surgery (LSS) [8,9,10,11], procedural accuracy has yet to be extensively documented.

Additionally, investigations into LSS procedures employing intraoperative CT navigation have reported several advantages, including reduced operating time and decreased staffing requirements [12,13,14]. Significantly, there is a notable reduction in radiation exposure to medical staff [15]. Recently, there have been reports of PPS fixation techniques in the lateral decubitus position for trauma, such as diffuse idiopathic skeletal hyperostosis (DISH) [16]. Typically, the PPS is inserted in the prone position, but if feasible in the lateral decubitus position, it may offer alignment correction benefits. LLIF is also seen to potentially reduce operating time compared to procedures requiring position changes [17]. While the accuracy of PPS placement in the prone position has advanced with intraoperative CT navigation over the past decade, its precision in the lateral decubitus position remains under-researched, especially concerning the thoracic spine. This study aims to assess the accuracy of PPS placement retrospectively, using intraoperative CT navigation in the lateral position, for thoracic to lumbar spine conditions. We hope to provide insights into the benefits of using intraoperative CT navigation in LSS and expand our understanding of PPS placement in the lateral decubitus position. This information aims to guide surgical decisions and enhance patient safety in spine disease treatments.

## 2. Materials and Methods

The research protocol received approval from our university’s Institutional Review Board before the study commenced and adhered to the principles outlined in the Declaration of Helsinki. Patients, or their families, were informed about using patient data for the objectives (23R-143). Given the retrospective nature of this study, the need for informed consent was waived.

### 2.1. Included Patients

This study encompassed patients who underwent posterior instrumentation with PPS from the thoracic to lumbar regions, utilizing an intraoperative CT navigation system, between April 2019 and September 2023. Patient data were collected from medical records and analyzed, including patient-specific characteristics (age, sex, height, body weight, body mass index (BMI), and diagnosis), surgical-related characteristics (number of PPSs used and the level of PPS inserted), and PPS-specific characteristics (Ravi scale grades [Grade I: no deviation, Grade II: <2 mm, Grade III: 2–4 mm, Grade IV: >4 mm] and the direction of PPS deviation) [18] (Figure 1). If there was a discrepancy between the evaluators, that screw was counted as a deviation. In terms of grade, it was also counted as a severe deviation. Patients without post-operative CT analyses were omitted from the study. The upper thoracic spine introduces particular anatomical and procedural challenges that can influence the accuracy and safety of PPS placement. Due to safety considerations and potential challenges in ensuring secure placements, we excluded patients with upper thoracic spine trauma from the LSS criteria.

The patient demographic details are shown in Table 1. Out of 170 patients, 93 patients (55%) underwent surgery at a single level, while 77 patients (45%) underwent surgery at two or more levels.

### 2.2. Surgical Technique

This study involved four seasoned spinal surgeons. No trainees participated. Three of these surgeons have over 20 years of experience since graduation and are certified spine surgeons by the Board of Directors of the Japanese Society for Spine Surgery and Related Research. In contrast, one surgeon has over 10 years of post-graduate experience but lacks the aforementioned certification.

This technique has been reported in a previous paper [11,12]. Briefly, patients were administered general anesthesia and placed in the lateral decubitus position on the surgical table. The choice of the lateral decubitus surgical approach for degenerative cases was to facilitate single-position LLIF. In trauma cases, the lateral approach was chosen for fracture reduction needs or due to thoracic trauma, thus reducing the risks associated with the prone position.

The patients were positioned in the lateral decubitus position and secured to the surgical table using adhesive tape. This is crucial to prevent the patient’s body from rotating, which could affect the alignment of the spine. 

Intraoperative CT navigation systems are vital for high-precision spinal surgeries. One of the first steps involves attaching a reference frame to the spinous process. Alternatively, a navigation reference pin can be inserted into the sacrum. The reference frame acts as a guide, ensuring that the PPSs are inserted at the exact planned location. Any misalignment or movement of this reference could lead to inaccurate placement, which, in spinal surgery, can have profound implications.

We employed the O-arm 2 (Medtronic plc, Dublin, Ireland) to capture 3D fluoroscopic images, which were transferred to the StealthStation surgical navigation system (S8; Medtronic Sofamor Danek, Minneapolis, MN, USA) for spinal navigation. Intraoperative CT navigation imaging was performed once for all cases. With the assistance of a computer-aided design derived from the navigation system, the PPS was ready to be inserted. The first step involved making a skin incision guided by virtual lines from the computer-aided design model. After this, an appropriate pilot hole was created using the Stealth-Midas system, another tool by Medtronic. This pilot hole served as a guide for the screw. Precision is vital, given the intricacies of spinal anatomy. A phenomenon known as “Skiving” can occur during screw placement, where the screw may deviate from its intended path. A meticulously crafted pilot hole can prevent this. Once the pilot hole is ready, the PPS was inserted with the help of the POWERASE driver (Medtronic Sofamor Danek) [21]. The PPS was inserted sequentially starting from near the reference frame. After the insertion of the PPS, we confirmed its position solely using fluoroscopy. In our study, the recorded operation time spanned the entire operation, covering both the LLIF procedure and the PPS placement (Figure 2).

### 2.3. Evaluation of the Accuracy of the PPS Placement

About 1–2 weeks after surgery, patients underwent a postoperative CT scan to assess PPS placement accuracy. We did not use intraoperative CT for post-surgical placement verification. Instead, we evaluated PPS positioning using postoperative 3 mm slice CT scans and a specific scoring system [19,20]. The best positioning options were the Ia and the Ib types. The IIa or IIb position must be evaluated for stability but usually does not require a revision. In the IIIa and IIIb malpositions, depending on stability or possible neurological irritation, a screw revision must be considered. This classification system describes the screw placement in the pedicle with particular focus on medial, lateral deviation from the optimal position.

### 2.4. Statistical Analysis

Statistical analyses were performed using IBM SPSS Statistics (version 23.0; IBM Corp., Armonk, NY, USA). All values are expressed as mean ± standard deviation. The Shapiro–Wilk test was used to confirm the normality of the data distribution. For the primary analysis, Student’s *t*-test or the Mann–Whitney U test was used to compare the two groups. Student’s *t*-test was used to analyze normally distributed data and the Mann–Whitney U test for non-normally distributed data. Comparisons between groups for categorical variables were assessed using the chi-squared test. The significance of the obtained results was judged at the 5% level.

## 3. Results

The data regarding PPS deviations are shown in Table 2.

A total of 836 PPSs were placed. Based on the Ravi scale, 21 PPSs (2.5%) were found to deviate (grades II to IV). The thoracic spine showed a significantly higher deviation rate at 7.4% (7 out of 94), compared to 1.9% (14 out of 742) in the lumbar spine.

In terms of perioperative complications, three patients had issues related to intraoperative CT navigation; one experienced a wound infection; two faced lower limb paralysis from PPS placement (excluding post-operative neurological deficits by LLIF); and one patient died perioperatively. This patient was a 77-year-old female with a history of interstitial pneumonia and rheumatoid arthritis, requiring steroid medication. She underwent an aortic dissection ascending replacement procedure. Following the surgery, she developed a pseudoaneurysm at the anastomosis site which was then corrected by cardiovascular surgery. The day after the surgery, she went into circulatory failure, necessitating a re-thoracotomy. The cause of bleeding remained unidentified until intercostal artery embolization was performed through interventional radiology to stop the hemorrhage. At that time, a CT scan revealed a DISH fracture, leading to a posterior fusion procedure from T6 to T12. After various surgeries, she suffered from a massive retroperitoneal hemorrhage, disseminated intravascular coagulation, and multi-organ failure, leading to her unfortunate passing.

Additionally, six patients required reoperation, which included cases involving re-decompression at the same level or PPS replacement.

The highest level of percutaneous instrumentation of the series was T6 and the lowest was L5 (Figure 3).

A total of 436 upside PPSs were inserted, and among them, 10 deviated (2.2%). On the downside PPSs, a total of 400 PPSs were inserted, and 11 of them deviated (2.8%). There was no statistically significant difference in screw deviation rates between upside and downside PPSs. Furthermore, we conducted a comparison between the thoracic and lumbar spine, but in both cases, there was no statistically significant difference in screw deviation rates between upside and downside PPSs (Table 3).

All PPSs had been designated as either upside or downside PPSs. We investigated the screw diameter and length of the PPSs (Table 4). There was no statistically significant difference in screw diameter between the thoracic and lumbar spine (6.5 ± 0.4 vs. 6.6 ± 0.3 mm, *p* = 0.137). However, screw length was longer in the lumbar spine (44.3 ± 4.6 vs. 45.7 ± 3.7 mm, *p* = 0.005).

When we measured the pedicle diameter, we found that the lumbar spine had a greater diameter than the thoracic spine (6.7 ± 1.9 vs. 10.8 ± 3.0 mm, *p* < 0.001).

We investigated the insertion location of the PPSs using a rating scale (Table 5). Two patients with 2b inner position abnormalities (two screws) were identified in the thoracic spine, occurring at T7 and T12. In contrast, we observed two patients (three screws) in the lumbar spine, with one at L3 and two at L5.

Furthermore, among patients with 3b inner position abnormalities, we found two patients (three screws) in the thoracic spine, one at T6 and two at T11. There were no deviations in the lumbar spine for the patients with 3b. The most common deviation pattern was type 1b, observed in 9 screws in total (1.1%) (Figure 4).

## 4. Discussion

We have found that improved PPS insertion accuracy can also be achieved by utilizing an intraoperative CT navigation system in the lateral position. Previous research has primarily focused on the precision of PPS placement and clinical outcomes in the prone position [18,22,23,24]. The accuracy of PPS insertion in the lateral decubitus position still needs to be elucidated. 

Though a few studies have tackled the accuracy of PPS placement in the lateral position [9,12,25], they are notably fewer than those examining the prone position. A previous fluoroscopy study reported a deviation rate of 5.1% during PPS insertion in the lateral decubitus position [26]. In contrast, Ouchida et al. [9] reported a reduction in deviation rates when using intraoperative CT navigation, with 1.8% in the lateral decubitus position and 4.0% in the prone position, highlighting its usefulness. While these data appear promising, our previous findings have concluded that the primary benefit of lateral decubitus PPSs using intraoperative CT navigation lies not only in the accuracy of PPS insertion but also in preventing facet joint violation [12]. 

Moreover, emerging innovations, like robotic spinal surgery and the growing popularity of LLIF, emphasize the potential of PPS insertion in the lateral decubitus position [27,28,29]. The LLIF approach is versatile and suitable for various degenerative conditions. It offers both sagittal and coronal deformity correction for adult spinal deformity and can address canal stenosis in lumbar degenerative disease through indirect decompression [30,31,32,33,34].

Previous research has reported thoracic PPS misplacement rates ranging from 4.2% to 25.7% in the prone position [35,36,37,38]. These data were derived when patients were prone, casting doubt on their applicability in the lateral position. Therefore, we investigated the accuracy of PPS insertion from the thoracic to the lumbar spine using intraoperative CT navigation. Our study also indicated a comparable accuracy rate for lateral decubitus PPS insertion of 7.4%, which aligns with previous findings obtained in the prone position. Additionally, it suggested that the inaccuracy rate of PPS placement is higher in the thoracic spine (7.4%) compared to the lumbar spine (1.9%). So far, the narrowest pedicles have been observed at the T3–T5 level, and there is considerable variability in the angles between the transverse pedicle axes and the PPSs. Moreover, it has been noted that the risk of screw malposition is significantly higher in the upper thoracic spine compared to the lower thoracic spine [35]. Our research did not include lateral decubitus PPS placements in this upper thoracic spine, potentially influencing the accuracy of our findings. If a PPS was introduced into this region in a lateral position, the accuracy could be worse than our reported 7.4%. Several factors might account for the diminished screw accuracy in the thoracic spine compared to the lumbar spine. The mismatch between the relatively large screw sizes, narrow pedicle diameter, unique thoracic spine anatomy, and the inclusion of only trauma cases in the thoracic category could be contributing factors. Given the variation in pedicle diameter between thoracic and lumbar spines [39] and potential trauma-induced factors, more research is essential in understanding these discrepancies fully.

We also investigated whether there was a difference in screw insertion accuracy between upside and downside PPSs. It has been reported that patients in the lateral position have difficulty accessing the inferior PPS insertion angle due to the limited working space between operating tables [40]. However, similar to previous reports [24,41], we found no difference in the PPS insertion accuracy between upside and downside PPSs. While fluoroscopy might complicate downside PPS insertion, intraoperative CT navigation could ameliorate this challenge.

Intraoperative CT navigation brings significant precision to surgical procedures, but its adoption is challenging. The initial investment in this technology can be substantial, requiring equipment, training, and maintenance resources, which may strain hospital budgets. Another concern is the potential increase in patient radiation exposure due to repeated intraoperative CT scans [15,42,43]. It is vital to weigh the benefits of improved navigation against the need to reduce radiation risks.

Most commonly, we observed type 1b deviations. The prevalence of such deviations might be influenced by the surgeons’ propensity to evade deviations into the spinal canal, mitigating potential neurological risks.

This study has several limitations. Firstly, it was confined to a single center, which may affect the generalizability of our findings to other institutions with different surgical practices and patient demographics. Secondly, the study’s retrospective design may introduce selection bias, affecting our control over data collection and interpreting causal relationships. Thirdly, the limited sample size could constrain the wide applicability of our findings. Due to this limited sample size, we require a greater number of inaccurately placed screws to analyze the risk factors influencing PPS accuracy effectively. These risk factors include patient attributes such as BMI, surgical variables like the positioning of the reference frame, and mismatches between the pedicle diameter and screw size. Fourth, while trauma patients were included, we did not detail the nature and severity of their traumas, which might influence PPS insertion accuracy. Fifth, we excluded cases of PPS placement in the upper thoracic spine in a lateral decubitus position, limiting our findings’ relevance to this anatomical area. Lastly, our primary focus was PPS placement precision, without an extensive examination of long-term clinical outcomes in the lateral position. Larger multi-center studies are recommended for more comprehensive insights in future LSS research. Additionally, prospective, longitudinal research can help assess the long-term outcomes of patients with PPS placement deviations, tracking complications, pain, and functionality over extended periods. This would enhance the study’s generalizability and depth.

## 5. Conclusions

In conclusion, our study highlights the difficulties in achieving precise PPS placement using LSS, especially in the thoracic spine. The inaccuracy rate of 7.4% in the thoracic spine, compared to 1.9% in the lumbar spine, clearly indicates the need for improvement in this surgical approach for the thoracic spine. Ongoing research is crucial to better the PPS insertion techniques in the lateral decubitus position, particularly for the thoracic spine. By addressing these challenges and investing in further research, we can enhance the accuracy of PPS placement and subsequently improve patient outcomes in spine surgery.

## Figures and Tables

**Figure 1 jcm-12-06914-f001:**
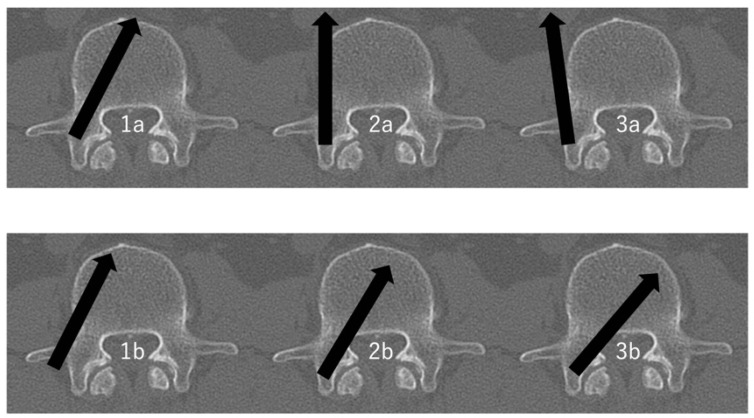
The evaluation of PPS positioning within the pedicle [19,20]. 1a: >Half of pedicle screw diameter within the pedicle and >half of pedicle screw diameter within the vertebral body. 1b: >Half of pedicle screw diameter lateral outside the pedicle and >half of pedicle screw diameter within the vertebral body. 2a: >Half of pedicle screw diameter within the pedicle and >half of pedicle screw diameter lateral outside the vertebral body. 2b: >Half of pedicle screw diameter within the pedicle and tip of pedicle screw crossing the middle line of the vertebral body. 3a: >Half of pedicle screw diameter lateral outside the pedicle and >half of pedicle screw diameter lateral outside the vertebral body. 3b: >Half of pedicle screw diameter medial outside the pedicle and tip of pedicle screw crossing the middle line of the vertebral body. Arrows indicate the direction of screw insertion.

**Figure 2 jcm-12-06914-f002:**
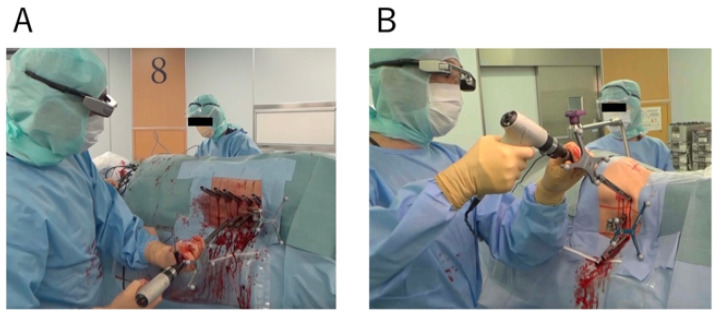
PPS insertion in lateral decubitus position ((**A**); Thoracic spine. (**B**); Lumbar spine).

**Figure 3 jcm-12-06914-f003:**
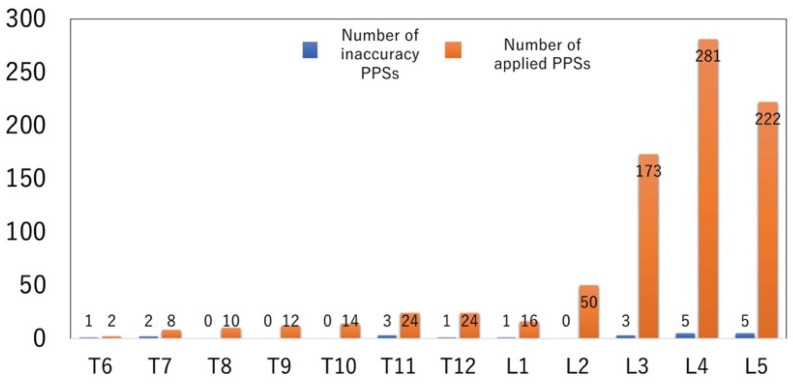
Number of applied PPSs related to the level of insertion (*n* = 836, highest T6, lowest L5).

**Figure 4 jcm-12-06914-f004:**
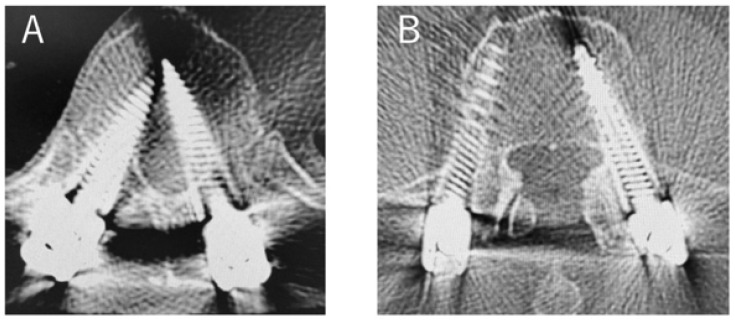
Case of PPS insertion deviation ((**A**); 3b with deviation in the insertion of PPSs on the left T11. (**B**); 1b with deviation in the insertion of PPSs on the right L4).

**Table 1 jcm-12-06914-t001:** Baseline characteristics of patients.

Characteristic		Value
No. of patients, *n* (LLIF/Trauma etc.)		170 (151/19)
No. of screws, *n* (LLIF/Trauma etc.)		836 (690/146)
Age (years), mean (SD)		71.0 (11.6)
Sex (male/female), *n*		94/76
Height (cm), mean (SD)		158.9 (9.8)
Body weight (kg), mean (SD)		61.1 (11.5)
BMI (kg/m^2^), mean (SD)		24.1 (3.5)
Indications *n* (%)	LCS + (LDS)	124 (73)
	DLS	11 (7)
	FS	8 (5)
	DISH Fx	9 (5)
	Thoracolumbar Fx	8 (5)
	LDH	6 (4)
	Synovial cyst	2 (1)
	Spondylitis	2 (1)
Number of levels operated *n* (%)	1 level	93 (55)
	2 level	60 (35)
	3 level	7 (4)
	4 level	7 (4)
	5 level	1 (1)
	6 level	2 (1)
	Mean	1.6 (0.9)

Data presented as mean (SD) or number of patients (%). BMI, Body mass index; LCS, Lumbar canal stenosis; LDS, Lumbar degenerative spondylolisthesis; DLS, Degenerative lumbar scoliosis; FS, Foraminal stenosis; DISH, diffuse idiopathic skeletal hyperostosis; LDH, Lumbar disc herniation.

**Table 2 jcm-12-06914-t002:** Screw accuracy, perioperative data, and complications. OR, operation; EBL, estimated blood loss; *p* < 0.05; * Statistically significant.

Characteristic		Value		
Screw Accuracy, Number (%)				*p* (II–IV)
Thoracic spine *n* (%)	Grade I		87 (92.6)	Thoracic spine vs. Lumbar spine 0.006 *
	Grade II		5 (5.3)	
	Grade III		2 (2.1)	
	Grade IV		0 (0)	
	Grade II–IV		7 (7.4)	
	Total		94	
Lumbar spine *n* (%)	Grade I		728 (98.1)	
	Grade II		5 (0.7)	
	Grade III		7 (0.9)	
	Grade IV		2 (0.3)	
	Grade II–IV		14 (1.9)	
	Total		742	
Thoracic ± lumbar spine *n* (%)	Grade I		815 (97.4)	
	Grade II		10 (1.2)	
	Grade III		9 (1.1)	
	Grade IV		2 (0.2)	
	Grade II–IV		21 (2.5)	
	Total		836	
Perioperative data, mean (SD)				
OR time (min)		116.6 (36.1)		
EBL (mL)		101.2 (117.0)		
Complications, number (%)				
iCT-related problems		3 (1.8)		
Surgical site infections		1 (0.6)		
Postoperative neurological deficit (PPS-related)		2 (1.2)		
Mortality		1 (0.6)		
Reoperation		6 (3.5)		

**Table 3 jcm-12-06914-t003:** Summary of accuracy data between upside and downside PPS.

Characteristic		Value		
Screw Accuracy, Number (%)		Upside PPS	Downside PPS	*p* (II–IV)
Thoracic spine *n* (%)	Grade I	44 (93.6)	43 (91.5)	
	Grade II	2(4.3)	3(6.4)	
	Grade III	1 (2.1)	1 (2.1)	
	Grade IV	0 (0)	0 (0)	
	Grade II–IV	3 (6.4)	4 (8.5)	1.000
	Total	47	47	
Lumbar spine *n* (%)	Grade I	382 (98.2)	346 (98.0)	
	Grade II	3 (0.8)	2 (0.6)	
	Grade III	4 (1.0)	3 (0.8)	
	Grade IV	0 (0)	2 (0.6)	
	Grade II–IV	7 (1.8)	7 (2.0)	1.000
	Total	389	353	
Thoracic ± lumbar spine *n* (%)	Grade I	426 (97.7)	389 (97.3)	
	Grade II	5 (1.1)	5 (1.3)	
	Grade III	5 (1.1)	4 (1.0)	
	Grade IV	0 (0)	2 (0.5)	
	Grade II–IV	10 (2.2)	11 (2.8)	0.826
	Total	436	400	

PPS; percutaneous pedicle screw. *p* < 0.05 indicated significant difference.

**Table 4 jcm-12-06914-t004:** Summary of screw size and pedicle diameter data.

Characteristic	Value			*p*
Screw Size, *n* (%)	Thoracic	Lumbar	Total	
5.5 (mm)	2 (2.1)	5 (0.7)	7 (0.8)	
6.0 (mm)	8 (8.5)	28 (3.8)	36 (4.4)	
6.5 (mm)	74 (78.7)	560 (75.1)	634 (75.5)	
7.0 (mm)	0 (0)	124 (16.9)	124 (15.0)	
7.5 (mm)	10 (10.6)	25 (3.4)	35 (4.2)	
Screw diameter, mean (SD)	6.5 (0.4)	6.6 (0.3)	6.6 (0.3)	0.137
35 (mm)	6 (6.4)	4 (0.5)	10 (1.2)	
40 (mm)	30 (31.9)	145 (19.5)	175 (20.9)	
45 (mm)	30 (31.9)	347 (46.8)	377 (45.1)	
50 (mm)	28 (29.8)	240 (32.3)	268 (32.1)	
55 (mm)	0 (0)	6 (0.8)	6 (0.7)	
Screw length, mean (SD) (mm)	44.3 (4.6)	45.7 (3.7)	45.5 (3.9)	0.005
Total	94	742	836	
Pedicle diameter (mm) mean (SD)	6.7 (1.9)	10.8 (3.0)	10.3 (3.2)	<0.001

*n* = number.

**Table 5 jcm-12-06914-t005:** Postoperative CT scan to evaluate the position of PPS.

	1a	1b	2a	2b	3a	3b	Inferior	ALL
Thoracic spine, *n*	87	2	0	2	0	3	0	94
(%)	92.6	2.1	0	2.1	0	3.2	0	100
Lumbar spine, *n*	728	7	0	3	3	0	1	742
(%)	98.1	0.9	0	0.4	0.4	0	0.1	100
ALL, *n*	815	9	0	5	3	3	1	836
(%)	97.5	1.1	0	0.6	0.4	0.4	0.1	100

*n* = number.

## Data Availability

The data presented in this study are available on request from the corresponding author. The data are not publicly available due to privacy or ethical restrictions.

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
