# Peer review of "Assessing Procedural Accuracy in Lateral Spine Surgery: A Retrospective Analysis of Percutaneous Pedicle Screw Placement with Intraoperative CT Navigation"

_jcm, 2023, doi:10.3390/jcm12216914_

Round 1

Reviewer 1 Report

Comments and Suggestions for Authors

Dear Authors,

Firstly, I would like to commend you on the comprehensive work and efforts put into "Assessing Procedural Accuracy in Lateral Spine Surgery: A Retrospective Analysis of Percutaneous Pedicle Screw Placement with Intraoperative CT Navigation". The topic is of critical importance in enhancing outcomes of lateral spine surgery and patient safety.

Abstract Feedback:

Comment 1

Consider specifying the exact methodology of the intraoperative CT navigation system usage.

Comment 2

The differentiation between the deviation rates of the thoracic and lumbar spine is noteworthy. Some clarity or hypothesization on the underlying reasons would be enriching.

Comment 3

A brief insight into any associated complications or comorbidities during the study, if observed, would be beneficial.

Comment 4

PPS Indication:

The abstract rightly highlights the importance of PPS in spine conditions. However, expanding on the indications of PPS – for instance, which specific conditions benefit the most, or in which scenarios is its use most critical – would be beneficial for a comprehensive understanding. A brief exploration into the clinical decision-making process surrounding the utilization of PPS might add a layer of depth to your study.

Introduction

Comment 1:

The importance of PPS in treating spine diseases is well-presented. Consider briefly highlighting the major complications from incorrect PPS placements to underscore the procedure's significance.

Comment 2:

While mentioning the transition from fluoroscopy to CT navigation, a short nod to any direct benefits to the patient (e.g., reduced radiation) might further strengthen the argument for newer techniques.

Comment 3:

You've identified a gap regarding PPS placement in the lateral decubitus position. To further emphasize this study's importance, briefly mention why understanding accuracy in this specific position is crucial.

Comment 4:

The study's objective is clear. A hint towards the expected findings or primary hypotheses could provide readers with a roadmap for the study.

Methods

Comment 1:

In the "Included Patients" section, it would be beneficial to provide clear inclusion and exclusion criteria. Were there any specific reasons some patients were excluded?

Comment 2:

Within the "Surgical Technique" section, mentioning the average years of experience or particular expertise of the surgeons can emphasize their proficiency.

Comment 3:

While the PPS insertion procedure is well-described, a brief explanation about the significance of each step might be helpful for the general readership.

Comment 4:

In the "Evaluation of Accuracy of PPS Placement" section, providing more context or explanations on the significance of types Ia, Ib, IIa, IIb, IIIa, and IIIb can clarify their importance in the assessment process.

Comment 5:

For the "Statistical Analysis" section, consider elaborating on the rationale for choosing specific tests or adding any post-hoc tests that were performed.

Results

Comment 1:

While you provided the overall deviation rates, it might be more informative to break down the deviations by grade (II, III, and IV) for both thoracic and lumbar spine, which can give more insight into the severity of the deviations.

Comment 2:

Clarifying the nature and impact of the issues related to intraoperative CT navigation may be beneficial, especially given the study's emphasis on the utility of this technology.

Comment 3:

Considering the serious nature of the complications, discussing the factors leading to lower limb paralysis in the context of PPS placement and explaining more about the one patient who died (aside from the multiple organ failure due to DISH fracture) might be valuable for the reader.

Comment 4:

When discussing deviations between upside and downside PPSs, elaborating on potential implications or reasons for these deviations, if known, would add depth to the analysis.

Comment 5:

The difference in screw lengths between the thoracic and lumbar spine is mentioned, but it might be helpful to provide context or a potential reason as to why this might be the case.

Comment 6:

For the pedicle diameter measurement, you noted a significant difference between the thoracic and lumbar spine. Offering an explanation or citing relevant literature might be useful to provide context for this difference.

Comment 7:

When discussing the insertion location abnormalities, offering insights or hypotheses on the clinical implications or reasons behind these abnormalities would make the analysis more comprehensive.

Comment 8:

The last statement notes that type 1b was the most common deviation pattern. Adding a brief discussion or hypothesis on why this particular pattern is the most common would be beneficial.

Discussion

Comment 1:

Topic: Contextual Comparison

Recommendation: Delve deeper into the clinical implications of the differences between the lateral decubitus and prone positions. In which clinical situations would one position be preferred over the other? Clarifying this could offer readers a comprehensive understanding of why positioning is vital and its direct impact on patient outcomes.

Comment 2:

Topic: Study Limitations

Praise: Recognizing the limitations of your study boosts the credibility and transparency of your research.

Recommendation: Consider specifying potential solutions or workarounds to the stated limitations in future research. For example, would multicenter studies help address the single-center limitation?

Comment 3:

Topic: Trauma Patient Information

Recommendation: While you've acknowledged the inclusion of trauma patients, the details about their specific injuries or trauma severity could be significant. This data might influence PPS insertion accuracy and might be valuable for the reader's comprehension.

Comment 4:

Topic: Upper Thoracic Spine Exclusion

Recommendation: Given that you excluded cases of lateral decubitus PPS placement in the upper thoracic spine, it may be beneficial to explain why. Were there specific challenges or risks associated with this area?

Comment 5:

Topic: Long-Term Clinical Outcomes

Recommendation: The research primarily focuses on PPS placement precision but doesn't extensively discuss the long-term clinical outcomes associated with the lateral decubitus position. Including or referencing a study about these outcomes could provide a more comprehensive perspective on the subject's importance.

Comment 6:

Topic: Future Research

Recommendation: Clearly outline steps or measures for future studies in this area. Highlight the importance of considering the mentioned limitations and propose a roadmap that future researchers could follow to further refine and expand upon your findings.

I hope this helps in making your research even more robust!

Comments on the Quality of English Language

Overall Quality: The manuscript is largely well-written and demonstrates a good command of the English language.

Clarity & Flow: The paper's flow and structure are logical, with each section providing clear information and building upon the previous one. However, some sentences could be streamlined for brevity and clarity.

Technical Terms: The use of technical terms is appropriate for the subject matter. It's recommended to ensure that all terminologies are consistent throughout the manuscript.

Recommendations:

In certain sections, there are phrases that could be made more concise to enhance readability.

Consider having a native English speaker or a professional language editing service review the manuscript to fine-tune any minor nuances or idiomatic expressions.

Conclusion: The manuscript, in its current state, is very comprehensible and adheres to the scientific standards of English language use. With a few minor adjustments, it would be ready for an international journal.

Reviewer 2 Report

Comments and Suggestions for Authors

Performing PPS insertion in the lateral decubitus position is a well-recognized method for reducing surgical time, commonly employed by surgeons. The authors of this study have conducted a comprehensive analysis of this technique, which proves to be valuable for spinal specialists engaged in similar procedures. I find the overall structure of the article well-organized, requiring only minor adjustments.

1. Being experienced in similar procedures, I've noticed that your surgeries proceed quite rapidly. Could you please clarify whether the calculation of OR time exclusively encompasses PPS insertion or also includes the preceding LLIF procedure? As some surgeons conduct both procedures simultaneously, including this detail would enhance the manuscript.

2. Could you specify the timing of the postoperative axial 3-mm slice CT scans? Were these scans conducted during the surgery or after the patients had returned to the recovery area? The text mentions two patients requiring reoperation due to PPS-related issues. With the availability of O-arm equipment in your operating room, was it utilized post-surgery to validate the accuracy of pedicle screw placement?

Round 2

Reviewer 1 Report

Comments and Suggestions for Authors

Congratulations! 

Comments on the Quality of English Language

English is well written except for the Discussion part. Maybe minor editing needed. 
